∂ | Open Peer Review | Clinical Microbiology | Research Article

# Reliability of CMV-IgG kinetics in the diagnosis of CMV primary infection: sensitivity, specificity, and clinical implications

Vincent Portet Sulla,[1,2] Rana Rafek,[1] Isabelle Bertin-Jung,[3] Jean-Pascal Siest,[3] Elise Bouthry,[4] Olivier Rogier,[1] Abir Jadoui,[1] Christelle Vauloup-Fellous,[1,2] Claire Perillaud-Dubois[5]

**ABSTRACT** Diagnosis of cytomegalovirus (CMV) primary infection (PI) during pregnancy relies on serology (CMV-IgG, IgM, and IgG avidity). However, as for toxoplasmosis, subsequent serology testing 3–5 weeks later is often performed to confirm the diagnosis. In this study, we aimed to show that testing CMV-IgG with different assays may lead to misinterpretation of CMV-IgG kinetics and to determine the sensitivity and specificity of CMV-IgG stability and significant increase to exclude or confirm recent CMV PI. We conducted a retrospective study on (i) a CMV-IgG external quality control program (2015–2022) and (ii) on CMV serology results obtained in our virology laboratory (2013–2023) in pregnant women with positive CMV-IgM and a subsequent serum sample collected 3–5 weeks later. Analysis of 21 CMV-IgG external quality control serum samples highlighted significant differences in CMV-IgG values, with variations up to a factor of 185 between different immunoassays for the same positive sample. In 434 pregnant women, the sensitivity of a significant CMV-IgG increase to predict recent PI was 32.9% (95% CI = 26.5–39.2), while CMV-IgG stability specificity to exclude PI <3 months was 32.9% (95% CI = 26.5–39.2). Our observations highlight the discrepancies in CMV-IgG values with different assays and the major importance of CMV-IgG avidity in the diagnosis of recent CMV PI in case of positive CMV-IgM. We also demonstrate that retesting IgG on a sample collected 3–5 weeks later is not helpful and can be confusing.

**IMPORTANCE** This article is the first to address cytomegalovirus (CMV)-IgG kinetics and their reliability in the serological diagnosis of CMV. In our experience, many clinical virologists and laboratory practitioners still rely on kinetics for diagnosis. However, our study clearly demonstrates that this approach is misleading and that avidity testing should always be performed. Additionally, we conducted a robust study highlighting discrepancies between CMV serology techniques, emphasizing the importance for practitioners, particularly gynecologists, to avoid monitoring serology results using different testing methods.

**KEYWORDS** CMV, serology, IgM, avidity, pregnancy, IgG

Cytomegalovirus (CMV) is the most frequent worldwide cause of congenital viral infection, affecting 0.5%–1% of all live births. Congenital CMV is a major cause of sensorineural hearing loss and intellectual disability (1–3). CMV can be transmitted to the fetus after primary or secondary maternal CMV infection, with a similar proportion of symptoms and sequelae in both situations (4–7). At birth, 13% of congenitally infected neonates are symptomatic with CMV-specific symptoms, including growth restriction, microcephaly, ventriculomegaly, chorioretinitis, sensorineural hearing loss, hepatitis, thrombocytopenia, and a purpuric skin eruption (2–6, 8). Risk of long-term sequelae is higher if CMV transmission occurs in the first or second trimester of pregnancy or during the peri-conceptional period (9–11). In immunocompetent patients, CMV

**Peer Reviewer** Inna Sekirov, British Columbia Centre for Disease Control, Vancouver, Canada

Address correspondence to Vincent Portet Sulla, vincent.portetsulla@aphp.fr.

The authors declare no conflict of interest.

primary infection (PI) is often asymptomatic. When symptomatic (8%–10% of cases), PI is usually responsible for a mild disease. Signs most frequently reported are isolated fever, asthenia, mononucleosis syndrome with cervical lymphadenopathy, and/or cytolytic hepatitis (12).

Diagnosis of CMV PI during pregnancy mainly relies on serology: detection of specific CMV-IgG and IgM, associated with CMV-IgG avidity in case of positive CMV-IgM (13, 14). For CMV-IgG avidity, specificity and sensitivity comprised between 90% and 100% depending on the assay (15–18). However, issues arise as CMV-IgG avidity is not available in all laboratories, and clinicians often rely on a positive CMV-IgM result or significant increase in CMV-IgG values 3–5 weeks later to diagnose CMV primary infection (as is usual for toxoplasmosis). However, this subsequent serology might not be performed in the same laboratory (and not by the same assay), leading to misdiagnosis. Moreover, in addition to positive IgM, a significant increase in CMV-IgG can be due to another cause other than PI, particularly polyclonal nonspecific stimulation of the immune system. Finally, it has been noticed that CMV-IgG may reach the plateau very quickly and appear stable even a few weeks after the onset of PI.

In our retrospective cohort study, we aimed (i) to highlight the discrepancies in CMV-IgG values obtained with different assays to show that testing CMV-IgG with different assays may lead to misinterpretation of CMV-IgG kinetics and (ii) to determine whether stable or increased levels of CMV-IgG are sensitive and specific markers to exclude or confirm recent CMV PI during pregnancy.

## MATERIALS AND METHODS

### External quality assessment data collection

Per French recommendations, all laboratories analyzing clinical samples need a specific accreditation (Norme NF EN ISO 15189). Consequently, it is mandatory that these laboratories participate regularly in external quality assessment (EQA) programs organized by national associations. One of these organizations (Biologie Prospective, France) provides participants (136–179 laboratories in France and countries nearby) with four quality control materials per year to assess CMV-IgG detection quality. These quality control materials, plasma or serum usually collected from a single donor, are tested routinely by laboratories, and results are reported to Biologie Prospective. Between 2015 and 2022, 21 positive quality control samples from the EQA program for CMV-IgG testing were tested by participants. Each sample was analyzed by 94–146 participating laboratories (Table 1; Table S1), allowing comparison of mean CMV-IgG values obtained with eight commercial immunoassays commonly used in France: Elecsys (Cobas e411, e601, e602, e402, and e801) (Roche Diagnostics, Germany), Immulite 2000 (Siemens HealthCare, Germany), Architect, Alinity i (Abbott Diagnostics), Vidas (bioMerieux, France), Liaison (Diasorin, Italy), and Access (Beckman-Coulter).

### CMV serology data collection for CMV-IgG kinetics analysis

In our laboratory (Hôpitaux Universitaires Paris Saclay), CMV serology (both IgG and IgM) is performed on serum collected from asymptomatic pregnant women during the first trimester of pregnancy (systematic screening). Moreover, we also test samples (from pregnant women) collected in one of the laboratories belonging to our network (systematic screening in other centers) and referred for further analysis to our laboratory due to positive CMV-IgM. All CMV serologic results (IgG, IgM, and IgG avidity in case of positive IgG) from pregnant women in our laboratory between January 2013 and December 2023 were retrospectively analyzed. Samples were included if both CMV-IgG and CMV-IgM were positive and if a second serum sample collected 3–5 weeks later for the same patient was available and CMV-IgM was also positive on the second sample.

**TABLE 1** Inter-assay variability in CMV-IgG titers: data from the external quality control program[a]

| EQA sample name | LIAISON CMV IgG II Mean | IMMULITE 2000 CMV IgG Mean | Elecsys CMV IgG (Cobas e411, e601, e602) Mean | Elecsys CMV IgG (Cobas e402, e801) Mean | Architect CMV IgG Mean | Alinity i CMV IgG Mean | VIDAS CMV IgG Mean | Access CMV IgG Cytomegalovirus Antibody Mean | Ratio of highest value to lowest |
|---|---|---|---|---|---|---|---|---|---|
| 2015-1 | 144.4 | 13.5 | 2,409.4 | | 392.4 | | 107.2 | 343.5 | 178.5 |
| 2015-2 | 118.8 | 12.6 | 96.108 | | 182.2 | | 68.2 | 379.6 | 30.0 |
| 2015-3 | 51.7 | 8.7 | 15.3 | | 75.3 | | 27.7 | 165.2 | 19.0 |
| 2016-2 | 85.7 | 9.1 | 154.7 | | 208.0 | | 51.5 | 549.5 | 60.5 |
| 2016-3 | 133.0 | 19.1 | 828.4 | | 253.1 | **152.48** | | 467.7 | 43.4 |
| 2016-4 | 35.4 | 8.8 | 3.9 | | 63.0 | | 20.5 | 221.2 | 56.8 |
| 2017-1 | 127.3 | 13.6 | 2,516.4 | | **343.89** | | 104.9 | 361.9 | 184.7 |
| 2017-2 | 105.1 | 12.4 | 73.2 | | 200.2 | | 71.6 | 406.9 | 32.8 |
| 2017-4 | 52.5 | 8.8 | 15.3 | | 84.6 | | 24.4 | 173.7 | 19.7 |
| 2018-2 | 90.0 | 13.9 | 366.8 | | 233.9 | 353.6 | 44.1 | 420.3 | 30.2 |
| 2018-4 | 19.1 | 2.1 | 2.8 | | 34.3 | 27.0 | 6.0 | 16.087 | 16.2 |
| 2019-1 | 53.2 | 8.9 | 16.2 | | 92.8 | 101.4 | 27.0 | 70.0 | 11.4 |
| 2019-2 | 135.9 | 15.3 | 2,339.4 | | **357.67** | 372.3 | 93.9 | 310.3 | 153.3 |
| 2019-4 | 103.9 | 10.9 | 430.7 | | 174.7 | 166.0 | 60.9 | 242.7 | 39.4 |
| 2020-1 | 21.1 | 2.2 | 2.8 | | 33.8 | 36.0 | 6.0 | 16.6 | 12.8 |
| 2020-3 | 22.8 | 3.8 | 163.3 | | 31.3 | 31.6 | 13.0 | 53.0 | 12.6 |
| 2020-4 | 84.1 | 8.4 | 68.2 | | 109.6 | 123.9 | 40.9 | 49.5 | 3.0 |
| 2021-3 | 59.2 | 8.3 | 1.6 | | 47.6 | 46.6 | 17.5 | 52.9 | 38.1 |
| 2021-4 | 141.4 | 18.9 | 2,120.9 | 2,215.2 | **522.74** | **519.98** | 73.7 | 442.5 | 30.0 |
| 2022-1 | 138.9 | 25.03 | 2,674.4 | 2,837.7 | 527.5 | 529.2 | 96.7 | 446.8 | 29.3 |
| 2022-4 | 132.2 | 21.5 | 503.0 | 474.4 | 236.4 | 258.9 | 106.7 | 414.4 | 4.7 |

[a]This table shows the variability of CMV-IgG titers measured using different commercial immunoassays as part of an EQA program. For each sample, the mean CMV-IgG values are reported. The data illustrate significant discrepancies between assays, highlighting the impact of assay selection on CMV-IgG quantification and the potential implications for clinical interpretation. Only assays used by at least five participants were included in the analysis (excluded data shown in gray cells). Reagents with a coefficient of variation (CV) greater than 20% were excluded from analysis; corresponding values are indicated in bold. Units: U/mL (Cobas, Liaison), AU/mL (Architect, Alinity, Access, and Vidas), and S/CO (Immulite).

## CMV serology assays and interpretation

CMV-IgG and CMV-IgM were measured with LXL (DiaSorin, Saluggia, Italy). In case of positive CMV-IgG and CMV-IgM, we always perform LXL CMV-IgG avidity (14, 19, 20). An index > 0.40 excludes recent CMV PI. If below 0.40, VIDAS (bioMérieux, Craponne, France) CMV-IgG avidity is used and interpreted as follows:

- Positive CMV-IgG/positive CMV-IgM/high CMV-IgG avidity (LXL > 0.4, or LXL < 0.4 and VIDAS > 0.65): PI <3 months excluded.
- Positive CMV-IgG/positive CMV-IgM/low CMV-IgG avidity (LXL < 0.4 and VIDAS < 0.20): recent CMV PI <1 month confirmed (20).
- Positive CMV-IgG/positive CMV-IgM/low CMV-IgG avidity (LXL < 0.4 and 0.20 < VIDAS < 0.40): recent CMV PI (1–2 months) confirmed (20).
- Positive CMV-IgG/positive CMV-IgM/moderate CMV-IgG avidity (LXL < 0.4 and VIDAS > 0.40 but <0.65): recent CMV PI (2–3 months) not excluded. Patients with this serological profile were excluded from the study because the dating of PI is often questionable in this situation.

These thresholds used to define CMV PI <1 month and 1–2 months were derived from previously published data and are not explicitly recommended by the manufacturers (20). They were applied in this study based on prior validation in similar clinical contexts and to better stratify the timing of infection during pregnancy.

Significant increase of CMV-IgG value is defined as at least doubling on two samples collected 3–5 weeks apart. Otherwise, the IgG value is considered stable. The delay of 3–5 weeks is currently used in toxoplasmosis serology to evaluate the stability of IgG values. A recent evolutive infection (<2 months) is diagnosed if the values double a month

apart. This marker, although very useful in parasitic serology, is frequently misused for the interpretation of viral serologies.

## Statistical analysis for EQA data

Each laboratory is evaluated qualitatively and quantitatively. For qualitative evaluation, each laboratory result is compared to an assigned result, which corresponds to results obtained by the majority of participants after checking consistency with the results obtained by the Biologie Prospective's expert. The laboratory's result is deemed "pass" if it matches with the assigned result and "fail" if not (e.g., for an expected positive CMV-IgG: "pass" if the laboratory's result is "positive" and "fail" if the laboratory's result is "negative").

For quantitative evaluation, statistical analysis is performed if the qualitative result assigned is positive, with a ratio between the result and the positivity threshold value strictly higher than one (>1).

If the number of quantitative results by reagent is greater than or equal to 7, the statistical plan applied is based on the robust method involving Algorithm A (Algorithm A—Annex C of NF ISO 13528) (20) as follows: outliers are automatically excluded; robust means, which corresponds to the assigned value, and robust standard deviations are calculated.

If the number of quantitative results by reagent is strictly lower than 7, the statistical plan applied is an arithmetic plan. Arithmetic means and arithmetic standard deviations are calculated.

In both cases, assessment is based on the $z$-score, which is the difference between the value reported by the laboratory and the assigned value, corrected for standard deviation: $z$-scores < 2, acceptable; $z$-score, 2–3 doubtful (recommendation of close monitoring); and $z$-score > 3, unacceptable.

## Statistical analysis for CMV-IgG kinetics

We evaluated sensitivity and specificity with 95% confidence intervals of CMV-IgG increase or stability as an independent serological marker to confirm or exclude recent CMV PI. Our reference assay to date PI is CMV-IgG avidity.

## RESULTS

### External quality control program results for CMV-IgG (Biologie Prospective)

Between 2015 and 2022, laboratories participating in Biologie Prospective's CMV EQA program tested 32 quality control materials using their routine immunoassay. Twenty-one of these had a positive result assigned for CMV-IgG. Statistical information (mean, standard deviations, and coefficient of variation robust or non-robust) for these 21 quality control materials was collected (Table 1; Table S1).

Only assays used by at least five participants were included in this study. Thus, on 19 reagents, eight immunoassays were analyzed: Elecsys CMV IgG—Cobas e411, e601, e602, e402, and e801 (Roche Diagnostics, Germany), Immulite 2000 CMV IgG (Siemens HealthCare, Germany), Architect CMV IgG or Alinity i CMV IgG (Abbott Diagnostics), Vidas (bioMerieux, France), Liaison (Diasorin, Italy), or Access CMV IgG Cytomegalovirus Antibody (Beckman-Coulter).

All manufacturers, except Siemens HealthCare (Immulite 2000 CMV IgG), have determined the linearity range for their assay and recommend sample dilution if the value of the sample tested is above the upper linearity limit. The values obtained may, therefore, exceed the upper limit.

Values ranged from 19.1 to 144.4 AU/mL with Liaison assay, from 1.6 to 2,837 U/mL with Cobas assays, from 2.1 to 19.1 S/Co with Immulite assay, from 31.3 to 529.2 AU/mL with Architect and Alinity i assay, from 6.0 to 107.2 AU/mL with Vidas assay, and from 16.6 to 549.5 AU/mL for Access assay (Table 1; Table S1).

Coefficients of variation calculated from means and standard deviations for each reagent are usually <20% (or even <10%, depending on the reagent). Coefficients of variation >20% were observed for some reagents on some quality control materials, but these results were excluded from the analysis.

Means for each assay in each survey are plotted in Fig. 1. Overall, great disparities in results are observed depending on the assay used. Indeed, the factors calculated between the highest mean and the lowest mean (in comparison with the Liaison assay) showed a large variability ranging from 1.1 to 38.1 (Table 1; Table S1). These factors range from 3.0 to 184.7 when considering high and low means of all assays (including Immulite assay) per survey (Table 1; Table S1).

Finally, we compared the CMV-IgG values obtained by eight commercial assays, and we observed that for the same sample, values are not comparable between assays. These are random differences, and no trend can be identified concerning assays with higher or lower values (e.g., in sample 2016-4 Roche is at 3.9 U/mL compared to Access at 221.2 AU/mL, and conversely, in sample 2020-3, Roche is at 163.3 U/mL compared to Access at 53.0 AU/mL).

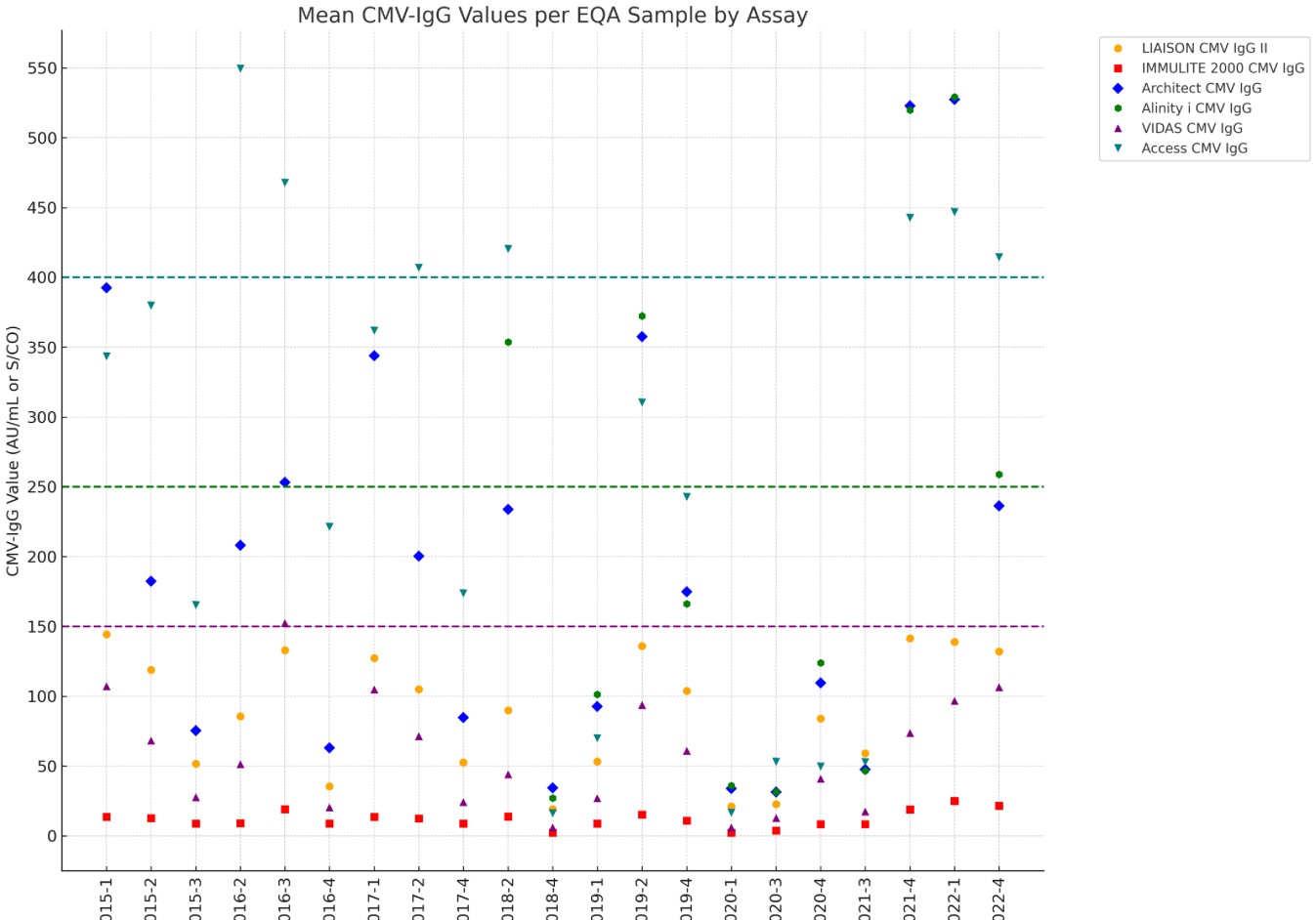

**FIG 1** IgG values obtained by the different assays used in EQA Samples (units: AU/mL for Architect, Alinity, VIDAS, Access, and Liaison; S/CO for Immulite). This figure shows the mean IgG values measured by different laboratories using various reagents and instruments in the EQA program. The y-axis represents values in AU/mL or S/CO, while the x-axis corresponds to the different EQA samples. Each point represents the mean value for a specific reagent. The upper linearity limits of each reagent are shown as horizontal dashed lines: VIDAS (150 AU/mL, purple dashed line), Access (400 AU/mL, teal dashed line), and Architect and Alinity (250 AU/mL, green dashed line). This figure visually illustrates the discrepancies in CMV-IgG values across different assays. For example, Abbott's Architect and Beckman-Coulter's Access systems frequently display variations exceeding a 100-fold difference, underscoring the critical impact of assay selection. These results emphasize the necessity of interpreting CMV-IgG kinetics with caution, particularly when sequential tests are conducted in different laboratories. Values from the Cobas Elecsys assay were not plotted to improve the visibility and graphical readability of the figure.

## CMV-IgG kinetics results in samples collected 3–5 weeks apart

Between 2013 and 2023, 9,798 serum samples collected from pregnant women during systematic screening were positive for CMV-IgM in our laboratory. All were tested for CMV-IgG and CMV-IgG avidity. Samples were included in our study only if

- CMV-IgG was positive (no seroconversion panel was included);
- VIDAS CMV-IgG avidity was either >0.65 (high avidity) or <0.4 (low and very low avidity);
- At least two serum samples from the same patient and collected 3–5 weeks apart were available.

Paired samples fulfilling these three conditions were available for 434 pregnant women (Fig. 2).

CMV-IgG avidity was high in 224/434 (51.6%) cases, allowing us to exclude recent CMV PI (<3 months). In this situation, a significant increase in CMV-IgG was observed in 5/224 (2.2%) cases, and CMV-IgG was stable in 219/224 (97.8%) cases. Therefore, in past infections, sensitivity of CMV-IgG stability to exclude a recent PI was 97.8% (95% CI = 95.8–99.7), while specificity was 32.9% (95% CI = 26.5–39.2).

CMV-IgG avidity was very low in 118/434 (27.2%) cases, allowing for confirmation of very recent CMV PI (<1 month). In this situation, a significant increase in CMV-IgG was observed in 60/118 (50.8%) cases, and CMV-IgG was stable in 58/118 (49.2%) cases. Therefore, less than 1 month after the onset of infection, the sensitivity of a significant CMV-IgG increase for predicting very recent PI was 50.8% (95% CI = 41.3–59.9).

CMV-IgG avidity was low in 92/434 (21.2%) cases, allowing for confirmation of recent CMV PI (1–2 months). In this situation, a significant increase in CMV-IgG was observed in 9/92 (9.8%) cases, and they were stable in 83/92 (90.2%) cases. To avoid misinterpretation of CMV-IgG value increase or stability, patients with IgG values over the assay's upper limit were excluded from analysis. CMV-IgG values of samples were spread across the full quantification range and not restricted to high-end plateau values. Therefore, 1–2 months after the onset of infection, the sensitivity of a significant CMV-IgG increase for predicting a recent PI was 9.8% (95% CI = 3.7–15.9).

Overall, if the onset of PI is less than 2 months, the sensitivity of a significant CMV-IgG increase for predicting recent infection is only 32.9% (95% CI = 26.5–39.2), and specificity is 97.8% (95% CI = 95.8–99.7).

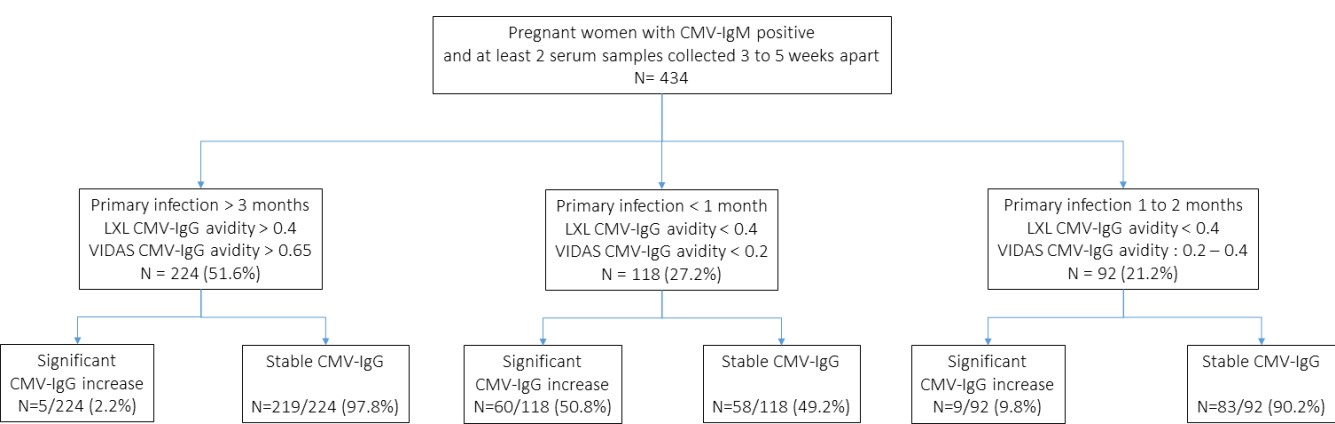

**FIG 2** Study population flowchart. According to CMV-IgG avidity results, pregnant women were divided into three groups: recent primary infection excluded (>3 months), very recent primary infection (<1 month), and recent primary infection (1–2 months). In each group, we evaluated the percentage of pregnant women having either significant CMV-IgG value increase or stable CMV-IgG values.

## DISCUSSION

Our observations highlight that CMV-IgG values may vary depending on the immunoassay. Immunity is a response to a complex set of antigens to which individuals may raise different levels of immune responses. Moreover, multiple strains of CMV circulate, increasing variability in immune responses between individuals (21). Finally, antigen preparations used in immunoassays differ among manufacturers, and characteristics of each assay vary in terms of solid phase, platform, detection system, standard, and interpretation range. Therefore, it is unlikely that all individuals' antibodies will react equally in all assays, causing obvious discrepancies. In our experience, clinicians often assume that CMV-IgG values (especially if expressed in IU/mL) are comparable between different assays/labs, which can lead to misdiagnosis and inappropriate patient management. As shown by the EQA program data analysis, when tests are performed in different laboratories using different assays, results may be inconsistent, leading to contradictory interpretations. As a result, to ensure accurate monitoring and consistency in serological monitoring, it is important to use the same assay/lab throughout pregnancy. Finally, quantification of CMV-IgG values could even be questionable, as having high levels of CMV-IgG has little clinical impact, and the patient will be considered immune, irrespective of the quantitative value.

Even though formal diagnosis of CMV PI is achieved with CMV-IgG seroconversion, as the "last" patient's seronegative serum is often unavailable, such documentation is rare. If CMV-IgG avidity assays are not easily available, CMV-IgM positivity and a significant increase in IgG values may be used as a diagnostic marker for CMV PI. Notably, the criterion for a significant increase in IgG values in our study—a doubling of the IgG value—mirrors the approach used in toxoplasmosis serology. In toxoplasmosis, this criterion is valid and provides a reliable diagnostic marker for recent infection. However, our results indicate that this is not applicable to CMV, and unfortunately, awareness concerning CMV-IgG kinetics early after infection onset is poor (17). Our study shows that a significant increase in CMV-IgG has very low sensitivity: 50.8% in very recent PI (<1 month) and 9.8% in recent PI (1–2 months). Moreover, we show that it may also be observed in 2.2% of cases in past infections (>3 months). Indeed, a significant increase in CMV-IgG can also occur in other situations, including infections with other pathogens associated with random polyclonal B cell stimulation or CMV non-primary infections (22, 23). Conversely, in very recent PI (<1 month), CMV-IgG stability is observed in 49.2% of cases and 90.2% in recent PI (1–2 months). However, our findings specifically apply to the DiaSorin LIAISON CMV-IgG assay used in this study. Other commercial assays could exhibit different performances regarding IgG kinetics, and further studies are needed to assess their diagnostic value in this context.

Overall, regardless of the CMV-IgG kinetics, CMV-IgG avidity is essential as low-avidity IgG is present only in early PI, increasing over 3–4 months to high avidity (24). Thus, CMV-IgG avidity has gained diagnostic importance in identifying CMV PI, and several commercial CMV-IgG avidity tests are currently available. Their performances to confirm a CMV PI range from 83% to 100% and from 71% to 100% (16, 25–31). All performances of the different strategies in the diagnosis of CMV PI are reported in Table 2.

Several groups reported substantial improvements in identifying at-risk pregnancies using diagnostic algorithms, including CMV-IgG avidity (11, 20, 24, 32–35). Indeed, it truly improves both accuracy and timeliness of CMV PI diagnosis, given that the positive predictive value of positive CMV-IgM and the sensitivity/specificity of CMV-IgG kinetics are quite poor (17, 32, 35, 36).

In patients with pre-existing CMV immunity (positive IgG), a significant rise in CMV-IgG (and high CMV-IgG avidity) and/or the presence of CMV-IgM often lead clinicians to suspect a CMV non-primary infection (NPI). Additionally, most clinicians will be reassured if CMV-IgG is stable. However, we have previously reported that a significant rise in CMV-IgG in NPI is observed in 18.9% of cases and that 81.1% of patients with NPI had stable CMV-IgG values (37). Finally, looking for variations of CMV-IgG values is useless either to assess PI or NPI.

TABLE 2  Diagnostic performances of CMV-IgM, IgG kinetics, and IgG avidity to diagnose CMV primary infection[a]

| Strategy | Performances |
|---|---|
| CMV-IgM to diagnose CMV PI | PPV of IgM = 16.4% if systematic screening during pregnancy (17) |
| | PPV of IgM = 36.7% in case of US abnormalities during pregnancy (17) |
| | PPV of IgM = 35.3% in case of clinical signs in the general population (17) |
| CMV-IgG kinetics | CMV-IgG increase (>2×) to confirm recent CMV PI: Sensitivity: 32.9%; specificity: 97.8% |
| | CMV-IgG stability to exclude recent CMV PI: Sensitivity: 97.8%; specificity: 32.9% |
| CMV-IgG avidity | Sensitivity to confirm CMV PI: 83%–100% (17) |
| | Specificity to exclude CMV PI: 71%–100% (17) |

[a]PPV, positive predictive value and US, ultrasound.

CMV screening during pregnancy is only recommended in a few countries (as in Italy, for example). Although not recommended by most European public health systems, it is more and more adopted by many general practitioners and obstetricians, especially since valaciclovir has proven its efficacy in first trimester CMV PI (37). Such screening provides an opportunity to identify seronegative women who can be counselled about adopting appropriate hygienic measures to prevent PI, especially in relation to their behavior with children, who are a major source of infection. It also allows early treatment to prevent mother-to-fetus transmission in case of maternal PI. Furthermore, screening aims to diagnose CMV PI early in pregnancy, allowing women to be referred to Reference Centers for appropriate management (antiviral treatment, close ultrasonography, amniocentesis, neonatal diagnosis, etc.) (35, 36). In clinical practice, in case of CMV-IgM positivity, our study provides clear evidence that CMV-IgG values, whether stable or increasing, are unreliable for dating infections. Only CMV-IgG avidity testing on the earliest available sample (with positive CMV-IgG) allows a reliable assessment, ensuring accurate patient management and avoiding unnecessary concern for pregnant women. This strategy also allows for a conclusion based on a single sample, eliminating unnecessary serological follow-up that might be misinterpreted and delaying the initiation of treatment, as well as unnecessarily worrying parents (as in most cases of positive CMV-IgM, CMV-IgG avidity is high) (17).

## Highlights

- CMV-IgG values in the same sample may vary up to 185-fold between different immunoassays.
- Sensitivity of significant CMV-IgG increase to predict recent primary infection is 32.9%.
- Specificity of CMV-IgG stability to confirm past infection is 32.9%.
- Retesting CMV-IgG 3–5 weeks later is usually unhelpful and may lead to misdiagnosis.
- CMV-IgG avidity testing remains essential for accurate diagnosis of recent CMV primary infection in pregnancy.

## ACKNOWLEDGMENTS

We thank the Biologie Prospective team for sharing their data and their valuable contribution to the analysis.

V.P.S. interpreted the study, performed formal analysis, and wrote the original draft. E.B. and R.R. performed formal analysis and reviewed and edited the manuscript. I.B.-J. performed formal analysis. J.-P.S. is the coordinator of Biologie Prospective's EQA programs. A.J. and O.R. performed serological analyses. C.V.-F. conceptualized the study and reviewed and edited the manuscript. C.P.D. conceptualized the study, designed the methodology, performed formal analysis, and supervised the study.

This work was a retrospective non-interventional study. Reclassification of biological remnants into research material was approved by the Institutional Review Board of the Assistance-Publique-Hôpitaux-de-Paris University Hospitals participating in the study. According to the French Public Health Code (CSPArtL.1121-1.1), such protocols are exempted from individual informed consent due to the retrospective chart review design and absence of identifying images or personal/clinical details that could compromise anonymity.

## AUTHOR AFFILIATIONS

[1]WHO Rubella National Reference Laboratory, Division of Virology, Department of Biology Genetics, Paul Brousse Hospital, Paris Saclay University Hospital, APHP, Villejuif, France
[2]Center for Immunology of Viral, Auto-immune, Hematological and Bacterial diseases (IMVA-HB/IDMIT), Paris Saclay University, INSERM U1184, CEA, Fontenay-aux-Roses, France
[3]Biologie Prospective, Villers les Nancy, France
[4]Department of Virology, Angers University Hospital, Angers, France
[5]Virology Department, Sorbonne University, Saint-Antoine Hospital, AP-HP, Pierre Louis Epidemiology and Public Health Institute (iPLESP), INSERM 1136, Paris, France

## AUTHOR ORCIDs

Vincent Portet Sulla  http://orcid.org/0009-0008-7929-0392

## AUTHOR CONTRIBUTIONS

Vincent Portet Sulla, Formal analysis, Investigation, Writing – original draft | Rana Rafek, Formal analysis, Writing – review and editing | Isabelle Bertin-Jung, Formal analysis | Jean-Pascal Siest, Resources | Elise Bouthry, Formal analysis, Writing – review and editing | Olivier Rogier, Investigation | Abir Jadoui, Investigation | Christelle Vauloup-Fellous, Conceptualization, Formal analysis, Methodology, Supervision | Claire Perillaud-Dubois, Conceptualization, Formal analysis, Methodology, Supervision

## DATA AVAILABILITY

Data will be provided by the authors on demand.

## ETHICS APPROVAL

As part of routine management in our hospital, patients are informed that their biological samples can be used for research unless they formally object. No patient has objected, and all data were analyzed anonymously.

## ADDITIONAL FILES

The following material is available online.

### Supplemental Material

**Table S1 (Spectrum00455-25-s0001.docx).** Inter-assay variability in CMV-IgG titers: data from the external quality control program (2015-2022).

### Open Peer Review

**PEER REVIEW HISTORY (review-history.pdf).** An accounting of the reviewer comments and feedback.

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
