## [Reviewer comments · Microbiology Spectrum]

Microbiology Spectrum

Reliability of CMV-IgG kinetics in the diagnosis of CMV primary infection: sensitivity, specificity and clinical implications

Vincent Portet Sulla, Rana Rafek, Isabelle BERTIN-JUNG, Jean-Pascal Siest, Elise Bouthry, Olivier Rogier, Abir Jadoui, Christelle Vauloup-Fellous, and Claire Perillaud-Dubois

Corresponding Author(s): Vincent Portet Sulla, Hopital Paul Brousse

Review Timeline:

Submission Date:	February 13, 2025
Editorial Decision:	April 1, 2025
Revision Received:	April 7, 2025
Editorial Decision:	April 25, 2025
Revision Received:	April 28, 2025
Accepted:	May 5, 2025

Editor: Alex Dulovic

Reviewer(s): Disclosure of reviewer identity is with reference to reviewer comments included in decision letter(s). The following individuals involved in review of your submission have agreed to reveal their identity: Inna Sekirov (Reviewer #2)

Transaction Report:

DOI: <https://doi.org/10.1128/spectrum.00455-25>

Re: Spectrum00455-25 (Reliability of CMV-IgG kinetics in the diagnosis of CMV primary infection: sensitivity, specificity and clinical implications)

Dear Dr. Vincent Portet Sulla:

Thank you for the privilege of reviewing your work. Below you will find my comments, instructions from the Spectrum editorial office, and the reviewer comments.

Following reviewer feedback, I invite you to submit a revised version of your manuscript.

Both reviewers were broadly supportive of the manuscript, but have made several suggestions which can be found below.

As a transferred manuscript, I appreciate that it was written and formatted with a different journal in mind. Please note that Microbiology Spectrum has different formatting requirements (which can be found at <https://journals.asm.org/journal/spectrum/article-types#research-articles>) which should be addressed during the revision.

Revision Guidelines

Sincerely,
Alex Dulovic
Editor
Microbiology Spectrum

Reviewer #1 (Comments for the Author):

In "Reliability of CMV-IgG kinetics in the diagnosis of CMV primary infection: sensitivity, specificity and clinical implications," Rafek and colleagues discuss the utility of cytomegalovirus (CMV) IgG titers in the diagnosis of primary infection. Primary CMV infection (PI) during pregnancy can result in congenital infection, which can cause long term neurologic sequelae and disability in impacted children. Accurate diagnosis of PI is needed to inform obstetric care. Current standard of care relies primarily on serology, including the detection of CMV-specific IgG and IgM, along with IgG avidity. While IgG avidity assays are highly specific and sensitive, they are not widely available and many laboratories rely on positive IgM titers or significant increases in IgG titer in two assays taken 3-5 weeks apart.

In this retrospective cohort study, the authors evaluated whether changes in IgG titer are a useful diagnostic marker while also highlighting the considerable variation in the results of different commercial assays. The authors demonstrated that titers were not comparable between assays by analyzing a set of standard samples on multiple platforms. While this is not surprising given the different characteristics of immunoassays, patient samples can be sent to different labs or run on different assays and clinicians may wrongfully assume that results expressed in the same units are comparable. This can lead to misdiagnosis and inappropriate patient management. The authors also show that CMV IgG titers are frequently stable and not diagnostically useful. Notably, in cases with low CMV-IgG avidity, which is only observed after recent PIs, significant changes in IgG titers were only observed in some cases.

The findings of this manuscript will help improve the diagnosis of primary CMV infections. The study is brief and may be more appropriate as an "Observation" than as a full research article. An "importance" section, which is required for either format, was not provided. Table 1 should be reformatted to improve readability without magnification.

Reviewer #2 (Comments for the Author):

Portet Sulla et al. report on the variability of CMV IgG commercially available assay outputs and on their laboratory's findings regarding CMV IgG dynamics during primary vs. non-primary infection, using DiaASorin assay outputs and comparing against the combination of DiaSorin and bioMerieux avidity test results as a reference standard.

The paper is interesting and informative from the perspective of highlighting the variability of commercial assay performance on proficiency standards and from the perspective of informing laboratories and clinicians regarding potential pitfalls of interpreting serial IgG test results with respect to infection timeline.

The paper could benefit from a number of edits:

1. "Titers" is used throughout to denote commercial assay outputs - none of them are truly titers and it would be better to choose an alternative term for them (e.g. IgG signals, IgG values or something along these lines). Along the same lines, it would be beneficial to include assay outputs (U/ml, S/CO, AU/ml) in Table 1.
2. Given that the authors highlight the differences in assay to assay performance, their conclusions regarding the pitfalls of serial CMV IgG measurements for timing the infection are realistically only truly valid with respect to DiaSorin assay performance. It is possible that there are other assays on the market that are able to have better PPV/NPV for primary vs non-primary CMV infection using serial measurements. This should be discussed.
3. Authors don't present any information on where the samples that were classified as "primary infection" by avidity testing but did not demonstrate a significant increase in IgG signals over time fell within the dynamic range of their assay. I.e. did the PI samples that did not have a rise in IgG fall on the extreme higher range of the assay quantification and had nowhere to rise, or did they fall throughout the assay range? This information should be presented and ideally performance of serial measures analyzed separately for samples that had "room" for rising vs. samples that maxed out on the assay range upfront.
4. Along the same lines it would be good to know whether CMV IgM testing was repeated on the 2nd serial sample in the dataset and whether the performance of serial IgG dynamics was dependent on either/both CMV IgM relative signals and presence/absence of CMV sero-reversion between samples. Even if CMV IgM wasn't tested on the 2nd sample, data can be analyzed relative to CMV IgM signals on the 1st sample.
5. Authors subdivide their samples into PI<1months and PI1-2months based on a combination of DiaSorin and BioMerieux avidity cut-offs proposed in reference 23. Are these cut-offs currently recommended by the manufacturer for interpretation of results? Or are they not currently manufacturer-approved? If they are currently manufacturer-recommended, this should be stated in the paper, in addition to providing the original reference for deriving the cut-offs. If they are not manufacturer-recommended, I think it would be better to stick with >3months vs. <3months classifications only for primary analysis and perhaps include these additional sub-divisions in supplemental materials.
6. Lines 260-263 - this section doesn't quite make sense. Significant rise in IgG and presence of IgM would typically make clinicians think of a Primary Infection, rather than NPI, as stated in first sentence.

Based on reference 37, 81.1% of patients with CMV NPI had stable IgG titers (Table 1) - it is >50%, but quite a bit over 50%, so not clear why 50% was chosen to quote.

Overall, based on reference 37, stable IgG levels do occur in most of NPIs, and I'm not sure what point is this paper being used to support or refute. Please clarify this section.

Minor comments:

1. In several places in the manuscript authors have IgG-CMV instead of CMV-IgG, which is used in most of the paper. Please

harmonize.

2. There are some minor grammatical/syntax errors throughout the manuscript that should be fixed (e.g. "diagnosis of CMV" rather than "diagnostic of CMV", "the same sample" rather than "a same sample", "in cases of positive CMV_IgM" rather than "in case of positive CMV IgM", "performed on serum" rather than "performed in serum", "lower than" rather than "lower to", etc.)
3. References need some formatting - e.g. ref. 40 is incomplete, ref 36 is duplicated
4. Line 254 "from 83 to 100% and from 71 to 100%" - based on Table 2 I understand that these refer to sensitivity and specificity respectively, but please specify this in the text as well.

Response to Reviewers

Reviewer #1

Comment 1:

« *The study is brief and may be more appropriate as an “Observation” than as a full research article* »

We respectfully thank the reviewer for this suggestion. However, we believe that the depth of analysis, the size of the cohort and the implications for clinical practice justify the current classification as a full research article. The study presents both a large retrospective cohort and a multi-assay external quality assessment dataset, supporting robust conclusions. We therefore kindly request to maintain the manuscript as a Research Article.

Comment 2:

« *An “Importance” section is required.* »

We have added an “Importance” section after the abstract as required.

Comment 3:

« *Table 1 should be reformatted to improve readability without magnification.* »

We thank the reviewer for this remark. To improve clarity and readability, we have split the data originally contained in Table 1 into two parts. A simplified version of **Table 1** is now included in the main manuscript. **Table 1** shows means of CMV-IgG values per assay and per EQA sample and ratio between the highest and the lowest values. The complete dataset, including the number of laboratories, standard deviations, and coefficients of variation (CV%), has been moved to **Supplementary Table 1** (Excel file). Units for each assay (U/mL, AU/mL, S/CO) have been explicitly indicated. The manuscript has been updated accordingly.

Reviewer #2

Comment 1:

« *The term “titers” is used throughout; suggest using “IgG signals”, “values” or similar terms.* »

We have replaced all instances of “titers” with the more appropriate terminology “IgG values”

Comment 2:

« *Conclusions regarding assay performance are only valid for DiaSorin.* »

This has been clarified in the Discussion section. We now explicitly state that our findings specifically apply to the DiaSorin LIAISON® CMV-IgG assay used in our study. Other commercial assays could perform differently with respect to CMV-IgG kinetics, and further studies are needed to assess their diagnostic value.

Comment 3:

« *Information is missing about where non-increasing samples fall within assay dynamic range.* »

We have included a clarification in the results section that these samples were distributed across the assay's quantification range and not clustered at upper limits. Furthermore, to avoid misinterpretation of CMV-IgG values increase or stability, patients with IgG values over assay's upper limit were excluded from analysis.

Comment 4:

« *Were CMV IgM tests repeated on second samples? Was there a relationship with IgM results?* »

CMV-IgM were systematically repeated on the second sample 3 to 5 weeks later and were also positive for all patients included. We added a sentence in Material and Methods section to clarify this point.

Comment 5:

« *Clarify if avidity thresholds are manufacturer-recommended or literature-derived.* »

We now state that the avidity thresholds are derived from the literature (reference 20) and not explicitly endorsed by the manufacturers.

Comment 6:

« *Section lines 260-263 are unclear; significant IgG rise and IgM presence usually indicate PI.* »

We agree that significant IgG rise associated with positive IgM first evoke a recent PI. But were we discussed this particular situation in presence of pre-existing CMV immunity (already known seropositivity). We specified this point in our discussion.

Comment 7:

« *Discrepancy between "more than 50%" and 81.1% – please clarify.* »

In a previous study, we reported that significant rise of CMV-IgG in NPI were observed in 18.9% of cases of NPI and that 81.1% of patients with NPI had stable CMV-IgG values. Comparing these CMV-IgG increase and stability rates with those observed in this study (respectively 32.9% of increase and 67.1% of stability in case of recent PI), we concluded that looking for variations of CMV-IgG values is useless either to assess PI nor NPI.

Minor comments:

« *Harmonize CMV-IgG terminology; correct grammatical/syntax issues and reference formatting.* »

We have reviewed and corrected all terminology inconsistencies (e.g., CMV-IgG vs. IgG-CMV), grammatical and syntax issues and reference formatting.

Re: Spectrum00455-25R1 (Reliability of CMV-IgG kinetics in the diagnosis of CMV primary infection: sensitivity, specificity and clinical implications)

Dear Dr. Vincent Portet Sulla:

Thank you for providing a revised version of your manuscript. Below you will find my comments, instructions from the Spectrum editorial office, and the reviewer comments.

Firstly, I wish to remind you that all reviewers are volunteers, colleagues of yours and should be treated respectfully. It is not acceptable for yourself or your colleagues to leave disrespectful comments about them within your submitted files. If you do have issues with a reviewer or feel that they have not understood your manuscript, you can always voice these concerns using professional language within the letter to the editor. Such tone and language within an open file is unacceptable and in contrast to the spirit of constructive peer review. I encourage you very strongly to be mindful of what you are submitting in the future.

Secondly, please note the reviewer comments listed below regarding Figure 1 and Table 1 that should be addressed in a further revision.

Revision Guidelines

Sincerely,
Alex Dulovic
Editor

Reviewer #2 (Comments for the Author):

The authors have addressed the comments raised in the first round of review with Microbiology Spectrum.

While reading through the revised manuscript and the review rebuttal, I noticed an inconsistency between Figure 1 and Table 1, which I didn't notice in the first round of review:

In Figure 1, all of the Diasorin XL IgG values hover somewhere below 50 on the Y-axis (as a separate note, the Y-axis doesn't list U/ml, which are the Diasorin reporting units per Table 1 legend; the legend of Figure 1 doesn't list Diasorin units either).

In Table 1, the Diasorin values for the same samples are reported to range between 19.1 and 144.4 (also as a separate note, the Table 1 legend still refers to CMV IgG Titers).

There are discrepancies between Figure 1 and Table 1 values for some of the samples tested on other platforms as well (e.g. 2021-4 on Alinity is just above 300 in Figure 1, yet 519.98 in Table 1; I don't see any datapoints above 100 for Vidas in Figure 1, yet there are several in Table 1, Architect sample 2015-3 is around 100 in Figure 1, but 75.3 in Table 1, etc.)

Either the Figure or the Table doesn't appear to be correct and has to be amended.

Response to Reviewers

Manuscript ID: Spectrum00455-25R1

Title: Reliability of CMV-IgG kinetics in the diagnosis of CMV primary infection: sensitivity, specificity and clinical implications

Reviewer #2:

We sincerely thank the reviewer for their careful assessment and constructive feedback.

First, we would like to sincerely apologize to the reviewer and the editorial office for the inappropriate comments that were inadvertently left in the submitted files. These comments originated from an earlier version of the manuscript prepared for submission to another journal and were mistakenly retained. Regardless of their origin, such remarks should not have appeared in our current submission, and we deeply regret this oversight. We have taken immediate measures to ensure that all future submissions are carefully reviewed to prevent any recurrence.

Regarding the scientific points raised, we acknowledge and apologize for the inconsistency between Figure 1 and Table 1 identified by the reviewer.

This inconsistency resulted from a plotting error during the preparation of Figure 1.

In response to the reviewer's comment, we have carefully corrected the issue :

- We fully regenerated Figure 1 using the correct mean CMV-IgG values extracted directly from Table 1.
- We corrected the units in the manuscript, Table 1, and Figure 1 to reflect that the LIAISON CMV IgG II assay reports in AU/mL and not U/mL.
- The Y-axis has been updated to indicate the correct units (AU/mL, or S/CO).
- The figure now consistently reflects the values reported in Table 1 across all assays.

Additionally, to improve the clarity and visibility of the figure, we decided not to plot the values obtained with the Cobas Elecsys assay. Including these data would have significantly compressed the Y-axis and reduced the readability of the comparative differences between other assays. We felt that removing Cobas Elecsys from the figure allowed a better graphical interpretation of inter-assay discrepancies without affecting the scientific conclusions of the study.

We sincerely thank the reviewer again for their valuable feedback, which helped us significantly improve the clarity and quality of our manuscript.

Overall changes :

- Updated Figure 1 to reflect the correct values and units.
- Corrected the description of units in Table 1 and throughout the manuscript (Liaison in AU/mL and not in U/mL).
- Corrected the description of units in the Supplemental Data legend to match the revised units (Liaison in AU/mL and not in U/mL).
- Revised the legend of Figure 1 to include the clarification that Cobas Elecsys values were not plotted for better visibility.

We remain fully available for any further adjustments if required.

Thank you very much for considering our revised manuscript.

Sincerely,

Dr. Vincent Portet Sulla

Re: Spectrum00455-25R2 (Reliability of CMV-IgG kinetics in the diagnosis of CMV primary infection: sensitivity, specificity and clinical implications)

Dear Dr. Vincent Portet Sulla:

Your manuscript has been accepted, and I am forwarding it to the ASM production staff for publication. Your paper will first be checked to make sure all elements meet the technical requirements. ASM staff will contact you if anything needs to be revised before copyediting and production can begin. Otherwise, you will be notified when your proofs are ready to be viewed.

Sincerely,
Alex Dulovic
Editor
Microbiology Spectrum